# The Future of Lipid-Lowering Therapy

**DOI:** 10.3390/jcm8071085

**Published:** 2019-07-23

**Authors:** Willemien van Zwol, Antoine Rimbert, Jan Albert Kuivenhoven

**Affiliations:** Department of Pediatrics, Section Molecular Genetics, University of Groningen, University Medical Centre Groningen, 9713 Groningen, The Netherlands

**Keywords:** atherosclerotic cardiovascular disease, residual risk, plasma lipids, Lp(a), Angptl3, ApoC-III, Omega 3 fatty acids

## Abstract

The recent introduction of inhibitors of proprotein convertase subtilisin/kexin 9 to lower low-density lipoprotein (LDL) cholesterol on top of statins or as monotherapy is rapidly changing the landscape of treatment of atherosclerotic cardiovascular disease (ASCVD). However, existing lipid-lowering drugs have little impact on lipoprotein(a) (Lp(a)) or plasma triglycerides, two other risk factors for ASCVD. This review summarizes the evidence and the rationale to target Lp(a) and triglycerides and provides an overview of currently tested strategies to lower Lp(a), apolipoprotein C-III and angiopoietin-like protein 3. In addition, it summarizes new findings on the use of omega-3 fatty acids (OM3FA) to fight ASCVD. With the exception of OM3FA supplementation, the promise of the experimental drugs discussed here depends on the long-term safety and efficacy of monoclonal antibodies and/or antisense oligonucleotides Clinical outcome trials will ultimately prove whether these new therapeutic modalities will reduce ASCVD risk.

## 1. Introduction

Even though mortality and morbidity from atherosclerotic cardiovascular disease (ASCVD) have been markedly reduced over the past decades, this reduction decelerates at a decreased rate and ASCVD remains the leading cause of death worldwide [1]. A central, modifiable causal risk factor in the development of atherosclerosis is dyslipidemia. To date, the focus of intervention is lowering plasma low-density lipoprotein (LDL) cholesterol levels with primarily statins, which are demonstrated to reduce major vascular events by 23% for every 38.7 mg/dL reduction [2,3]. Despite progress to lower LDL cholesterol in primary and secondary care, many patients remain at high ASCVD risk. This is illustrated by a recent European survey showing that only one-third of patients using lipid-lowering drugs reach desirable LDL cholesterol levels (<70 mg/dL) [4]. While adherence to evidence-based generic lipid-lowering medication remains a serious concern, a new class of efficacious drugs has recently emerged. Monoclonal antibodies against proprotein convertase subtilisin/kexin type 9 (PCSK9) reduce LDL cholesterol with 51% on top of statins or as monotherapy with an additional risk reduction of 15% [5,6]. Despite high costs, these drugs are currently becoming more widely available to patients. In the meantime, the research community and the pharmaceutical industry are focusing on possibilities to further reduce risk by targeting other risk factors. Mendelian randomization studies have in this regard recently established that elevated plasma levels of Lp(a) and triglycerides (TG) should be considered as independent causal risk factors for ASCVD [7,8,9]. These statistical analyses may be interpreted with care in view of the interrelationship between plasma lipids. Yet, these ideas are supported by observations that increased Lp(a) and TG levels increase risk despite low LDL cholesterol levels or statin treatment [10,11].

This review provides an update on recent data to reduce Lp(a) and to reduce TG through targeting hepatic production of apolipoprotein C-III (apoC-III) and angiopoietin-like protein 3 (ANGPTL3). It also summarizes the latest information on the use of omega 3 fatty acids (OM3FAs) to reduce ASCVD, compounds that not only affect TG but also inflammation (reviewed in Reference [12]). Figure 1, panel B, illustrates the pharmaceutical targets/interventions that will be discussed while Table 1 provides a summary of the main characteristics of the respective compounds.

## 2. LPA, Apolipoprotein (a), Lp(a)

### 2.1. Structure and Function

The *LPA* gene is expressed in the liver and encodes for apo(a). When covalently bound to apoB100 [21], the main protein component of LDL, it forms an LDL-like particle named Lp(a) (Figure 2). Apo(a) protein size can range from 200–800 kDa due to genetic copy number variation encoding for the kringle IV type 2 domain (KIV-2). The smaller isoforms are associated with increased plasma levels of Lp(a) [22,23]. Genetic variation of *LPA* is estimated to explain 91% of the variation in Lp(a) levels [24] and is only minimally impacted by dietary and lifestyle factors [25]. Interestingly, Lp(a) is also known as a major carrier of oxidized phospholipids (OxPL) in plasma [26,27]. Together with lipoprotein-associated phospholipase A2 (Lp-PLA2) these molecules stimulate pro-inflammatory pathways and plaque progression [28]. While the *LPA* gene is only found in a subset of primates and hedgehog [29], several animal models have provided insight into the mechanisms of how Lp(a) increases atherosclerosis [30]. Despite decades of research, the exact assembly, pathophysiology, and catabolism of Lp(a) remain enigmatic [31].

### 2.2. Observational

Several lines of evidence have implicated Lp(a) as a risk factor for ASCVD [8,32,33,34] and aortic valve stenosis [35,36]. A recent meta-analysis showed a statistically independent, almost linear relationship between plasma Lp(a) concentration and ASCVD risk in patients using statins [10]. In line with this, Wei et al. [37] showed that *LPA* gene variation is associated with coronary heart disease, independent of statin-induced change in LDL cholesterol [37]. Remarkably, Lp(a) has a stronger association with all-cause mortality than LDL cholesterol for a similar cholesterol content increase [33], implying that the effects of Lp(a) may not be explained by its cholesterol content alone. Although debated, a reduction of Lp(a) by 65.7 mg/dL was recently calculated to render the same effect on coronary heart disease reduction as an LDL cholesterol reduction of 38.7 mg/dL [38]. Some caution may be warranted here as these estimations are based on genetic epidemiological studies in which the actual cholesterol content of Lp(a) is not measured but estimated [39,40].

Following recommendations of the European Atherosclerosis Society Consensus Panel, Lp(a) levels should be <50 mg/dL, which is below the 80th percentile of the Danish Caucasian population [41]. Increased ASCVD risk could be expected at lower levels depending on the assay used and the population studied [42,43]. South Asians and Latin Americans present in this regard with higher Lp(a) concentrations and increased risk of myocardial infarction compared to Africans, Arabs, Chinese, Europeans and Southeast Asians [44]. Since 24% of 531,144 patients analyzed in the referral laboratory in the United States [45] and 46% of 247 patients with heterozygous familial hypercholesterolemia (FH) in Spain have plasma levels of Lp(a) >50 mg/dL [46], the number of individuals that could potentially benefit from treatment is evident.

Statins can increase Lp(a) levels by 9–20% arguing against increased LDL receptor (LDLR) mediated clearance of Lp(a) [47]. Increased *LPA* expression and apo(a) production association with statin-use is likely to explain this outcome [47]. On the other hand, drugs targeting PCSK9 (Evolocumab) or apoB (Mipomersen) have been shown to reduce Lp(a) plasma levels (reviewed by Tsimikas [23]). A 14% Lp(a) reduction in a study with Evolocumab did however not affect arterial wall inflammation [48] suggesting the need for greater Lp(a) reduction.

### 2.3. Drug Development

Experimental therapies reduce hepatic apo(a) synthesis through targeting its mRNA with antisense oligonucleotides (ASO). Following subcutaneous administration, these ASOs induce *LPA* mRNA degradation in the liver and thereby prevent protein synthesis. Several ASO-based strategies have been tested in the clinic with ASO conjugated to N-acetylgalactosamine currently being reported to be most efficient. This modification increases uptake by asialoglycoprotein receptors in hepatocytes. IONIS-APO(a)-L_RX_ is the first ligand conjugated antisense version and roughly 30 times more potent than IONIS-APO(a)_RX_ [13]_._

In Phase I trials, weekly application of IONIS-APO(a)-L_RX_ in individuals with elevated Lp(a), showed dose-dependent reductions of Lp(a) levels up to 92% [13]. This was seen with a concomitant decrease of apo(a) associated OxPL (72%), LDL cholesterol (20%) and apoB (19%). Of note, an almost complete down regulation reduces LDL cholesterol by only 20%.

A phase II trial with 20 mg AKCEA-APO(a)-L_RX_ weekly or 60 mg monthly showed that 98% and 81% of patients with established CVD and elevated Lp(a) levels achieved Lp(a) levels <50 mg/dL, respectively, without safety concerns [14]. A phase III cardiovascular-outcome study with AKCEA-APO(a)-L_RX_ is on the docket. Taken together, impressive progress has been made to lower Lp(a) levels but there is as of yet no evidence that reducing Lp(a) levels reduces ASCVD which would validate Lp(a) as a causal risk factor.

## 3. APOC3, apoC-III

### 3.1. Function

ApoC-III is primarily synthesized in hepatocytes and to a smaller extent in the intestine. Secreted into the circulation, it is present on high-density lipoproteins (HDL) and triglyceride-rich lipoproteins (TRL). TRL include chylomicrons, very-low-density lipoproteins (VLDL) and intermediate-density lipoproteins (IDL). It has initially been shown that overexpression of apoC-III in mice results in hypertriglyceridemia [49] whereas depletion of apoC-III decreases plasma TG levels and protects from postprandial hypertriglyceridemia [50]. Additional studies in mice have demonstrated that apoC-III attenuates TG hydrolysis through inhibiting lipoprotein lipase (LPL) and hepatic lipase (HL), and through increasing TG incorporation into VLDL in the liver (Figure 3) [51,52,53,54]. On the other hand, others have reported that apoC-III primarily affects plasma lipids through attenuating TRL uptake via members of LDL receptors but not Syndecan-1 (SDC1) [55,56,57]. This is supported by data showing that apoC-III interferes with the binding of apoB and apoE-containing lipoproteins to LDLR [58,59]. In addition, in vitro data suggest a direct role of apoC-III in endothelial cell activation and monocyte adherence [60] while it also plays a role in insulin resistance and β-cell function [61]. Despite extensive literature on apoC-III, its exact role in lipid metabolism remains to be fully elucidated [62], which is likely associated with the notion that it moves freely among lipoproteins.

### 3.2. Observational

Heterozygous carriers of gain-of-function (GOF) variants in the *APOC3* gene present with elevated plasma TG (32%) when compared to non-carrier relatives [63]. On the other hand, heterozygous carriers of *APOC3* loss-of-function (LOF) variants have lower TG (39%) and ASCVD risk (40%) [64]. Heterozygosity for *APOC3* LOF is associated with a higher conversion rate of VLDL to LDL with little effect on uptake compared to their siblings [65]. These authors postulate that increased conversion of VLDL would reduce ASCVD risk. Recently, four apparently healthy homozygotes for LOF variants with near-absent plasma apoC-III were shown to have slightly further reduced TG levels, markedly increased HDL cholesterol and similar LDL cholesterol levels compared to heterozygotes [66]. A decrease of 1 mg/dL apoC-III in plasma has been reported to be associated with a 4% risk reduction for CVD in models adjusted for age and sex [64].

### 3.3. Drug Development

The major strategy to reduce hepatic apoC-III production is via ASOs (Volanesorsen; ISIS 304801) [67] (For a description how ASOs work, see paragraph on drug development under Lp(a)). Volanesorsen was initially developed for familial chylomicronemia syndrome (FCS) and familial partial lipodystrophy (FPL), rare genetic disorders characterized by severe hypertriglyceridemia. Phase II trials with weekly subcutaneous injection of the highest dose of Volanesorsen (300 mg) as monotherapy, reduced apoC-III by 80% and plasma TG by 71%. These changes were accompanied by a 46% increase in HDL cholesterol, an 11% reduction of non-HDL cholesterol and an increase of 118% in LDL cholesterol [68]. The latter result is surprising when considering that congenital apoC-III deficiency does not affect LDL cholesterol levels [66]. Volanesorsen has also been reported to improve insulin sensitivity and is anticipated to be of use in patients with type 2 diabetes [69]. Phase III studies (APPROACH and COMPASS) with participants with FCS or severe hypertriglyceridemia (TG > 500 mg/dL), showed 84% reduction in apoC-III and >70% reduction in TG levels after weekly administration of 300 mg Volanesorsen [15,70]. Remarkably, the drug increased LDL cholesterol by 139% in FCS patients with low LDL cholesterol at baseline (28.2 mg/dL) while non-HDL cholesterol decreased [71]. It is unfortunate that the effects on LDL cholesterol in severe hypertriglyceridemic patients cannot be found in the public domain. The ongoing BROADEN study follows 60 FPL patients for 1–3 years with expected outcomes in 2019. The APPROACH Open Label Study is following FCS patients for 65 weeks and is estimated to be completed in 2020. Permission by the FDA for clinical use currently depends on the outcomes of ongoing clinical trials because Volanesorsen was found to increase the risk of thrombocytopenia in FCS patients.

A next-generation ligand-based ASO (ISIS 678354) renders similar effects as Volanesorsen without increasing LDL cholesterol. Phase I/II trials with 30 mg weekly administration resulted in 84% reduction in apoC-III, a 71% reduction in TG, a 56% increase in HDL cholesterol and a 17% decrease in LDL cholesterol [16]. Both multiple-dose groups, as well as weekly or monthly-dose groups, showed reduced LDL cholesterol. A phase II trial in patients with hypertriglyceridemia and established CVD will be completed this year [72].

## 4. ANGPTL3, Angiopoietin-Like Protein 3

### 4.1. Function

*ANGPTL3* is almost exclusively expressed in the liver and belongs to a gene family encoding for the angiopoietin family of secreted glycoproteins. ANGPTL3 is involved in regulating the distribution of TG between skeletal/heart muscle and adipose tissues where they are oxidized or stored, respectively. Following the initial finding that obese mice with a LOF mutation in *ANGPTL3* show hypolipidemia [73], the protein has been shown to inhibit LPL and endothelial lipase(EL)-mediated hydrolysis of plasma TG and HDL phospholipids, respectively (Figure 4) [67,73,74,75]. Extensive studies in mice have led to the conclusion that inactivation of ANGPTL3 (using antibodies) alters apoB-containing lipoproteins such that they are cleared more rapidly from the circulation without direct involvement op hepatic lipoprotein receptors [76]. The latter is interesting as it would offer an alternative to most drugs which are acting via the LDL receptor pathway. Antibodies against ANGPTL3 in a humanized mouse model has been shown to reduce plasma TG (84%), total plasma cholesterol (52%) and atherosclerotic lesion size [18].

### 4.2. Observational

Complete *ANGPTL3* deficiency in humans is very rare and causes familial combined hypolipidemia characterized by reduced plasma concentrations of TG (62%), LDL cholesterol (48%) and HDL cholesterol (46%) [77,78,79]. Homozygotes for LOF mutations, but not heterozygotes, are reported to have higher LPL activity, lower free fatty acids in plasma and increased insulin sensitivity compared to non-carriers [80]. Dewey et al. [18] showed that 246 heterozygotes for LOF variants had 27% lower TG, 9% lower LDL cholesterol but unchanged HDL cholesterol levels compared to 58,089 non-carriers, and a 41% reduced risk of coronary artery disease. Another meta-analysis of 19 studies with 21 different LOF variants exhibits a 34% lower risk of coronary artery disease [81].

### 4.3. Drug Development

Phase I and II trials with either monoclonal antibodies (Evinacumab) or ASOs (IONIS-ANGPTL3-L_RX_) are in early development [17,82,83]. It is possible that both therapies will render different outcomes as antibodies theoretically only block protein function in the circulation while ASO therapy blocks protein synthesis and thus also putative intracellular functions of the target protein.

In healthy adults with moderate hypertriglyceridemia or increased LDL cholesterol (>100 mg/dL), 20 mg/kg Evinacumab administration intravenously has been shown to reduce plasma TG by 50%, LDL cholesterol by 23% and HDL cholesterol up to 18% after 15 days (compared to placebo) [17]. Increased ANGTPL3 levels were observed and indicate target binding of Evinacumab. In an open-label study, Evinacumab was administered to 9 patients with homozygous FH which resulted in a mean reduction of 47% in TG, 49% in LDL cholesterol and 36% in HDL cholesterol [82], indicating that the drug can indeed reduce plasma lipids independent of the presence of functional LDL receptors. Heterozygous FH with persisting hypercholesterolemia despite maximal statin treatment and those at risk for acute pancreatitis are currently recruited for phase II trials (NCT03175367 and NCT03452228).

Phase I trials with weekly administration of 60 mg of IONIS-ANGPTL3-L_RX_ to healthy volunteers with hypertriglyceridemia resulted in reductions of ANGPTL3 (85%), TG (50%), LDL cholesterol (33%) and HDL cholesterol (27%) [83]. A phase II trial in patients with type 2 diabetes was completed in April 2019 and results are awaited.

Taken together, both monoclonal antibody therapy and ASO against ANGPTL3 are remarkably efficient with outcomes that at the highest dosages used mimic lipid and lipoprotein profiles of homozygotes for *ANGPTL3* LOF mutations.

## 5. Omega 3 Fatty Acids

### 5.1. Type and Function

To date, three OM3FA supplements are approved by the FDA for the treatment of severe hypertriglyceridemia (TG > 500 mg/dL). Despite decades of largely inconclusive clinical outcome studies, OM3FA is still suggested to have cardioprotective effects. Ongoing research focuses on the effective dose, and specific combinations of fatty acids and chemical form for optimal bioavailability [84,85,86]. Main types of OM3FA in oily fish include eicosapentaenoic acid (EPA), docosahexaenoic acid (DHA) and to a lesser extent, docosapentaenoic acid (DPA). Meta-analyses show that both EPA and DHA reduce TG with various effects on LDL and HDL cholesterol [87]. EPA is mostly studied and described to inhibit inflammation, cholesterol crystal formation, to reduce oxidative properties of atherogenic lipoproteins and to increase plaque instability (reviewed in References [88,89]). DPA can be metabolized to DHA and EPA [90] and also exhibits positive effects on cardio-metabolic parameters and improves lipid metabolism in mice [91,92]. DPA is less-extensively studied owing to the price and availability of the pure compound. Recent studies on DPA are summarized in Drouin et al. [93].

### 5.2. Observational

Widely available dietary supplements containing EPA and DHA are distinct from prescription drugs that require FDA safety approval prior to marketing (reviewed in Reference [94]). The three OM3FA-based drugs approved by the FDA for the treatment of severe hypertriglyceridemia are, (1) OM3FA ethyl esters (Lovaza/Omtryg; containing EPA and DHA); (2) icosapent ethyl (Vascepa; exclusively containing EPA); (3) omega-3 carboxylic acids (Epanova; containing a mixture of EPA, DHA and DPA).

As indicated above, OM3FAs are variably reported to have beneficial effects on plasma lipid metabolism but with little effect on ASCVD [95]. This was seen in a large meta-analysis (77,917 participants) [96], the recent ASCEND study (>15,000 patients with diabetes) [97] and the VITAL-study (a primary prevention trial with >25,000 participants) [98]. Surprisingly, major beneficial outcomes were recently observed in the REDUCE-IT trial: a reduction of 25% in cardiovascular mortality on top of the use of statins [20]. Oral intake of 4 g Vascepa per day for one year was shown to decrease plasma TG levels with 18%, with a slight increase of 3% in LDL cholesterol [20]. Based on these findings, the American Diabetes Association has decided to update the “Standards of Medical Care in Diabetes” and now considers Vascepa for diabetic patients with ASCVD or patients on statins with elevated TG to reduce cardiovascular risk [99]. It remains important to elucidate the molecular keys responsible for this success, which are likely not limited to only moderate lowering of TG [100].

### 5.3. Drug Development

The rather short-term EVAPORATE study evaluates Vascepa in 80 statin-treated patients with elevated TG (200–499 mg/dL) and follows progression of low-attenuation plaque volume, for 9–18 months [101]. Several long-term studies are also ongoing (STRENGTH, RESPECT-EPA, OMEMI), using different OM3FA composition and dosages (1.8–4 g), targeting patients on statins with hypertriglyceridemia (TG > 180 mg/dL), stable CAD or patients who suffered from myocardial infarction, respectively. The first long-term study will be completed in 2020 (reviewed in Reference [89]). MAT9001 (EPA and DPA) is in phase II trials, to support future FDA approval for severe hypertriglyceridemia [102]. Data from a study which compares MAT9001 with Vascepa is expected by the end of 2020 [102].

## 6. Future Perspectives

In a significant proportion of patients, LDL cholesterol reduction with statins and possibly ezetimibe and bile-acid sequestrants is insufficient to reduce the risk of ASCVD. Further LDL cholesterol reduction is possible with PCSK9 inhibition but high costs associated with the use of the currently registered monoclonal antibodies block general use thus far. The possible introduction of siRNAs(LIB003 [103]) or ASOs(SPC5001 [104]) against PCSK9, not discussed in this review, may provide some relief in the future. Another experimental drug, Bempedoic acid (FDA approval pending) that can lower LDL cholesterol up to 21% (discussed in References [105,106]), may also offer more room to further reduce LDL cholesterol on top of statins at, hopefully, lower costs. Combining lipid-lowering with anti-inflammatory compounds, which are under development, may also provide further risk reduction though the costs are likely to be an issue once again [107].

Currently available efficacious drugs to lower LDL cholesterol and the general downfall of therapies that target HDL (discussed in Reference [108]) have contributed to a shift to target elevated Lp(a) and TRL. Recent genetic support to target Lp(a) and TG as causal risk factors, combined with the technical advances to use monoclonal antibodies or antisense strategies has led to major advances in the development of new therapeutic modalities to possibly treat residual ASCVD risk (see Table 1; for comprehensive overview of existing and emerging drugs see Reference [109]). With these new means, targeting apo(a), apoC-III and ANGPTL3 in hepatocytes has been shown to be highly effective in numerous proof-of-concept studies. An initial concern that ASOs against apoC-III increased LDL cholesterol [68], appears to be solved with a new formulation [16] but the risk of thrombocytopenia remains an issue to be resolved. Important to the field of ASOs, this effect appears related to the target, apoC-III, and not the technology itself. The observed undesired reduction in HDL cholesterol following ANGTPL3 downregulation was anticipated when considering human genetic data. Fortunately, the latter data also show a reduced risk of ASCVD [18].

Despite apparent discomfort of subcutaneous injections or intravenous administration of ASOs or monoclonal antibodies, they offer advantages over daily intake of oral drugs (associated with compliance issues), especially when administration of once per month/season is within reach [110,111]. While these features are attractive, the ultimate general use of these compounds in the clinic is strongly dependent on costs and health benefits. As the costs of any new drug that reaches the market is likely to be high, attempts to individualize cardiovascular care are in this regard much needed. It is likely that the TG-lowering therapies discussed in this review will ultimately be used in patients that are at risk for ASCVD due to obesity, insulin-resistance and/or suffer from diabetes. These patients are now often treated with statins while their primary lipid-associated atherosclerotic risk is not directly related to increased LDL cholesterol but to elevated TG associated with low HDL cholesterol. It will in this regard be interesting to see whether TG-lowering drugs exert their effects through a reduction in TG or apoB which is suggested as a predictive marker [112,113]. In addition, patients with hypercholesterolemia as an apparent direct result of high Lp(a), may be offered targeted therapy. This requires, however, identification of such patients via Lp(a) measurements in the clinic.

The application of experimental drugs discussed in this review will naturally be dependent on results of the ongoing/planned outcome trials. Their ultimate application, when successful, will largely depend on the associated costs which may become manageable when personalized medicine will actually be developed and applied. On the other hand, the recent success of icosapent ethyl [20] shows that there are also other means to lower ASCVD risk that are maybe more affordable.

## Figures and Tables

**Figure 1 jcm-08-01085-f001:**
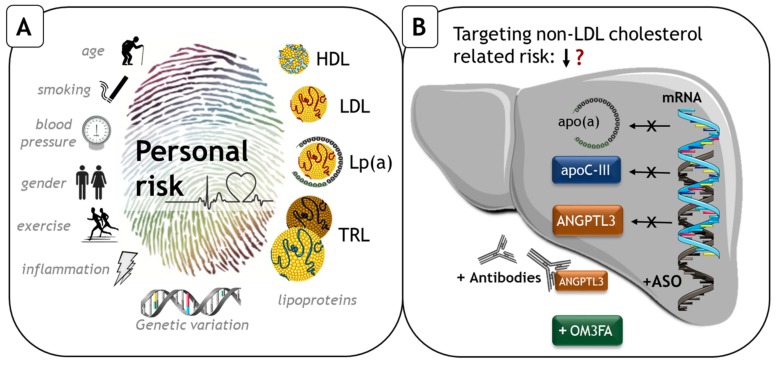
(**A**) Panel A summarizes the major factors associated with the risk of atherosclerotic cardiovascular disease (ASCVD). (**B**) Panel B shows strategies that are currently tested to reduce the risk beyond low-density lipoprotein (LDL) cholesterol lowering, which is the focus of this review. Apo(a), apoC-III and ANGPTL3 are all targeted with antisense oligonucleotides (ASO) which, after subcutaneous injections, bind to the respective mRNAs in the liver to inhibit protein synthesis. Following the successful introduction of monoclonal antibodies against PCSK9, antibodies against ANGPTL3 in the circulation are also tested. Finally, omega-3 fatty acids (OM3FA) are given as oral supplements to reduce triglycerides (TG) and inflammation. Abbreviations: HDL, high-density lipoprotein; LDL, low-density lipoprotein; Lp(a), lipoprotein (a); TRL, triglyceride-rich lipoproteins; mRNA, messenger RNA; Apo, apolipoprotein; OM3FA, omega-3 fatty acid.

**Figure 2 jcm-08-01085-f002:**
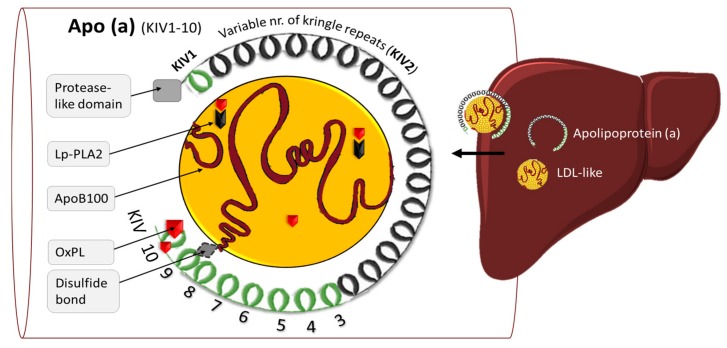
Schematic view of Lp(a). Abbreviations: apo(a), apolipoprotein (a); OxPL, oxidized phospholipids; Lp-PLA2, lipoprotein-associated phospholipase A2; KIV, kringle IV.

**Figure 3 jcm-08-01085-f003:**
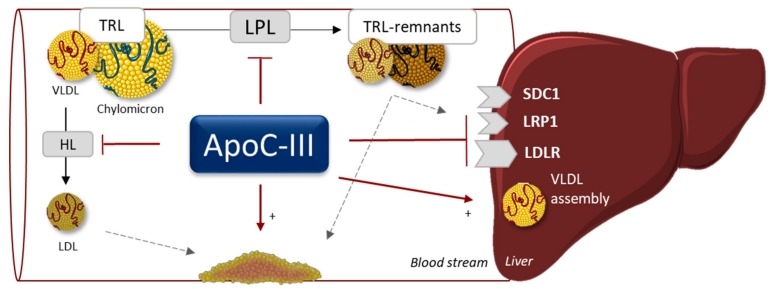
Apo-CIII promotes hypertriglyceridemia via different mechanisms. It inhibits lipolysis via inhibiting lipoprotein lipase (LPL) and hepatic lipase (HL) activities. It delays hepatic clearance of TRL and favors assembly and/or secretion of VLDL. Abbreviations: VLDL, very low-density lipoprotein; LDL, low-density lipoprotein; TRL, triglyceride-rich lipoprotein; SDC1, syndecan-1; LRP1, low-density lipoprotein receptor-related protein 1; LDLR, low-density lipoprotein receptor.

**Figure 4 jcm-08-01085-f004:**
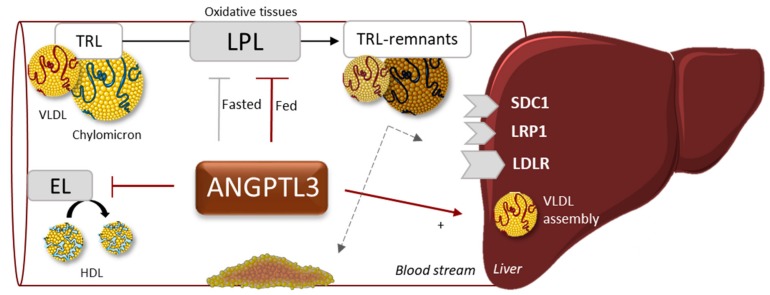
ANGPTL3 is reported to affect plasma TG through inhibition of lipoprotein lipase (LPL), especially in the fed state and HDL metabolism through inhibiting EL-mediated phospholipid hydrolysis. Last but not least, it also affects the assembly of VLDL in the liver. Abbreviations: EL, endothelial lipase; HDL, high-density lipoprotein; VLDL, very-low-density lipoprotein; TRL, triglyceride-rich lipoproteins; SDC1, syndecan-1; LRP1, low-density lipoprotein receptor-related protein 1; LDLR, low-density lipoprotein receptor.

**Table 1 jcm-08-01085-t001:** Targeting risk of plasma lipid-driven atherosclerotic cardiovascular disease beyond LDL cholesterol.

Target	Name of the Drug	Mechanism of Action	Lipid Effect		Stage of Development	Comments*
Lp(a)	1. IONIS-APO(a) RX 2. IONIS-APO(a)-LRX	ASO to block apo(a) synthesis and thereby Lp(a) formation	1. Lp(a) ↓ 72% Lp(a) ↓ 92%	[13]	Phase 2–3	Lp(a) levels are not measured in the clinic
				[14]		
APOC-III	Volanesorsen (AKCEA-APOCIIIRX)	ASO to block ApoC-III synthesis	TG ↓ 73%	[15]	Phase 3	Increases LDL cholesterol, thrombocytopenia risk
	AKCEA-APOCIII-LRX	ASO to block ApoC-III synthesis	TG ↓ 71%	[16]	Phase 2	
ANGPTL3	Evinacumab	Monoclonal antibodies target ANGPTL3 in plasma	TG ↓ 50%	[17]	Phase 3	Reduces HDL cholesterol
	IONIS-ANGPTL3-LRX	ASO to block ANGPTL3 synthesis	TG ↓ 50%	[18]	Phase 2	Reduces HDL cholesterol
OM3FA	1. OM3FA-EE/Lovaza 2. OM3FA-CA/Epanova 3. OM3FA-IPE/Vascepa	Reduced TG levels, inflammation, oxidative properties of atherogenic lipoproteins and increases plaque instability	1. TG ↓ 45% 2. TG ↓ 31% 3. TG ↓ 18–27%	[19,20]	Approved for HTG-patients	Inconclusive results on cardiovascular outcomes

* Potential drawbacks for all ASOs and monoclonal antibodies include discomfort and injection site reactions. Abbreviations: APO, apolipoprotein; ASO, antisense oligonucleotide; mRNA, messenger RNA; OM3FA, omega-3 fatty acid; EE, ethyl esters; CA, carboxylic acid; IPE, icosapent ethyl; HTG, hypertriglyceridemia.

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
