# Peer review of "The Future of Lipid-Lowering Therapy"

_jcm, 2019, doi:10.3390/jcm8071085_

Reviewer 1 Report

In the present review article entitled “The future of lipid-lowering therapy”, the authors summarize the available literature on the use of lipoprotein, apolipoprotein C-III and angiopoietin-like protein 3 inhibitors, and omega-3 fatty acids as novel treatment strategies for atherosclerotic cardiovascular disorder. They have discussed the function and clinical benefits of inhibitions of lipoproteins, ApoC-III and angiopoietin-like protein 3.  Overall, it is a well-written and up-to-date review article. However, I have the following comments:

Minor comments:

1.     The sentences are convoluted throughout the manuscript. The language needs some editing.

2.     There are many grammatical mistakes. The authors have used , (comma) and . (point) interchangeably (531.144 patients, line 96; 65.7 mg/dL, line 86).

Major comments:

1.    It would be better if the authors have provided a table showing the name of molecules, their beneficial and adverse effects along with the reference of the studies.

2.    They have not included any information on microsomal triglyceride transfer protein inhibitors and HDL mimetics. They should include a small section on these drugs as well.  

Author Response

Revision JCM 553343 entitled: ‘The future of lipid-lowering therapy. Targeting risk of plasma lipid-driven atherosclerotic cardiovascular disease beyond LDL cholesterol’.

Please see the attachment in which we have addressed each of the suggestions and comments of the reviewers. Our revised review is also given in the text below the comments, with major changes marked in blue.

Reviewer 2 Report

I really enjoyed reading this very interesting review. I think the Authors have summarized very well the wide and newest scenario of the lipid-lowering therapy, which represents a fundamental and essential weapon for the treatment of atherosclerotic cardiovascular diseases. The paper is clear and good prepared, and very explicating images are reported. Below, my considerations:

-        Line 74: There are some trials in progress to evaluate it?

-        Line 100: “Statins tend to increase Lp(a) levels”. Please explain the concept better and add a reference.

-        Line 199: How was the decrease of atherosclerosis in a humanized mouse model assessed?

-        Line 217-18: Please add a reference here.

-        Line 271-72: In which target vessel the progression of low-attenuation plaque volume was followed?

-        Line 280-324: Perhaps it would be better to separate the conclusions of the paper from the future perspectives, in order to better focus the review findings.

Author Response

(The authors gave the same response as above.)
